

# Data-driven emulation of melt ponds on Arctic sea ice

Simon Driscoll[1,5], Alberto Carrassi[2,1], Julien Brajard[3], Laurent Bertino[3], Einar Örn Ólason[3], Marc Bocquet[4], and Amos S. Lawless[1,5]

[1]University of Reading, The School of Mathematical, Physical and Computational Sciences (SMPCS), University of Reading, Reading, RG6 6BB, United Kingdom
[2]Department of Physics and Astronomy, University of Bologna, Viale Carlo Berti Pichat, 6/2, Bologna, 40127, Italy
[3]Nansen Environmental and Remote Sensing Center, Jahnebakken 3, Bergen, N-5007, Norway
[4]CEREA, École des Ponts and EDF R&D, Île-de-France, France
[5]National Centre for Earth Observation, Reading, UK

**Correspondence:** Simon Driscoll (s.driscoll@pgr.reading.ac.uk)

**Abstract.**

The genesis, development and disappearance of melt ponds on sea ice are complex, insufficiently understood, and driven by metre-scale mechanisms unseen by numerical models. Melt pond formation is thus parametrised with substantial uncertainty. Despite melt ponds playing a major role in sea ice thermodynamics, melt pond parametrisations have traditionally not been
included into GCMs, and for instance do not play a role in the IPCC assessments of when sea ice will disappear from the Arctic in summer.

Previous research showed it was possible to learn a state-of-the-art physical parametrisation of melt ponds (from model data) using neural networks. The emulator was suitable for use in a thermodynamic sea ice model (the Icepack model) and ran stably for about ten years. In this study, we follow on from that work and develop a novel data-driven parametrisation of melt
pond fraction. Using pan-Arctic satellite observations and reanalysis data we show that it is possible to learn and predict to a sufficient degree the melt pond fraction as determined from Medium Resolution Imaging Spectrometer (MERIS) and Ocean and Land Colour Instrument (OLCI) data, the target in our supervised learning setup, from well chosen observationally based inputs. Our deep learning emulator has been intentionally designed to be pointwise with the perspective of being suitable for incorporation within physical models of sea ice such as Icepack.

Our results prove the concept that it is possible to learn parametrisations directly from data for sea ice thermodynamical processes. In doing so our work provides a viable emulator of melt ponds for use in GCMs and demonstrates a route for developing and further refining observationally based data-driven emulators of melt ponds that are ready for implementation in GCMs. We also briefly discuss these future avenues for advancing on this work and further developing data-driven emulators of melt ponds.

Furthermore, the results show the difference in modelling melt ponds over different Arctic regions. In particular, it highlights the critical importance and need for better observations and understanding in the marginal ice zone which will be crucial for future impacts.



# 1 Introduction

Sea ice, one of the Earth system components most sensitive to climate change (Castellani et al., 2020), plays a fundamental role in the Earth's climate (e.g., Previdi et al., 2021; Sévellec et al., 2017; Dethloff et al., 2019), and climate models predict the possibility of ice-free summers occurring before or by the end of the 21st century (e.g. Wang and Overland, 2009; Boé et al., 2009). Losing high-albedo sea ice allows an increased absorption of solar radiation by the oceans, and this causes further surface warming (Manabe and Stouffer, 1980). Pistone et al. (2014) found that between 1979 and 2011 averaged over the globe the albedo change resulting from sea ice loss was equivalent to 25% of the direct forcing from $CO_2$ during the same period. This Arctic amplification is important for understanding the impacts of climate change (Hall, 2004; Previdi et al., 2021), and understanding the processes governing sea ice melt and ice-albedo feedbacks are crucial to predicting the future state of sea ice.

Each year surface melting occurs as part of the sea ice annual cycle and this snow and ice melt creates melt ponds that form over the surface of the Arctic sea ice. Melt ponds have a dramatic impact on the albedo of the ice surface: melt ponds may have an albedo as low as $0.15$, in contrast to the albedo of bare sea ice which can be up to 0.8 (Flocco et al., 2015). Melt pond properties are determined by a number of factors (Holland et al., 2012) that elude systematic description. Covering up to 50% of the surface of the sea ice (Flocco et al., 2015), their evolution on Arctic sea ice in summer is one of the main factors affecting sea ice albedo and hence the polar climate system (Li et al., 2020).

With sea ice reaching its maximum extent in March, melting processes over the melt season cause the sea ice to decline in an annual cycle where it reaches its minimum in September (Kwok, 2018). During this seasonal evolution the Arctic sea ice albedo undergoes multiple distinct phases starting with a uniformly high albedo surface, until pond formation and evolution from melting lead to a spatially heterogeneous ice surface, followed by freeze-up during Autumn (Perovich et al., 2002). Whilst the timing of melt onset varies by location (e.g., Zege et al., 2015), there exist similarities across Arctic locations in the melt process.

The development of parametrisations is necessary to account for the many phases and complexities of melt ponds. Numerical experiments with sea ice models demonstrate the sensitivity of sea ice thickness to melt pond parametrisations (e.g., Ebert and Curry, 1993; Driscoll et al., 2024), and models lacking these parametrisations can overestimate the thickness of the summer sea ice by up to 40% (Flocco et al., 2010, 2012). Moreover, their inclusion will likely become increasingly important for predicting climate change impacts, as melt ponds occur at a higher proportion over thinner ice (Feng et al., 2021).

In general, parametrisations are necessary for many climate processes that occur on scales smaller than the resolution of current models throughout the climate system. Examples not only include the formation and evolution of melt ponds on sea ice, but cloud formation processes, ocean vertical mixing and more (e.g., Lock, 1998; Pacanowski and Philander, 1981; McFarlane, 2011). Yet these parametrisations, often developed on heuristic assumptions, contribute substantially to uncertainty in climate projections (Flocco et al., 2012; Zanna et al., 2019; Yuval and O'Gorman, 2020). Interest in using machine learning (ML) to





emulate parametrised processes in climate simulations has grown considerably (e.g., Brenowitz and Bretherton, 2018; Rasp et al., 2018; Chantry et al., 2021; Finn et al., 2023), since the work of Krasnopolsky et al. (2013a). The case of melt ponds and the related surface albedo is particularly timely since recent progress in remote sensing provides a wealth of information (Istomina et al., 2020a), which is challenging for physically-based process models to match accurately.

In our previous work (Driscoll et al., 2024) we have shown the very large sensitivity of a state-of-the-art sea ice model (Icepack) and its physics-based parametrisation of melt ponds (the "level-ice" parametrisation) to the specification of their many tunable parameters. This sensitivity is not only very large, with simulated sea ice thickness over the Arctic Ocean region differing by multiple metres after only a decade due only to changes in these melt pond parameters, but it is also strongly temporally and spatially varying. This situation further motivated investigating the use of ML to discover new parametrisations.

Along this line, in Driscoll et al. (2024) we have shown that neural networks are able to "learn" the original physics-based parametrisation very accurately such that the incorporation of the ML-based parametrisation in the full sea ice model (in an online fashion) leads to skillful and stable predictions of key sea ice variables for very long time horizons (a decade).

    Moving one step forward, leveraging on observational data only, in this study we emulate the physical processes that cause the formation and evolution of melt ponds. Some related efforts exist in the very recent literature, yet with key differences.

In particular, as opposed to the work of Peng et al. (2022), we do not try to interpolate across a temporal series of melt pond observations, but instead to use satellite data of melt ponds as a target for our supervised learning, creating an emulator representing physical processes - and taking as input the physical variables that are available in a numerical forecast model. The emulator can then potentially be used for predicting future melt ponds as well being incorporated in sea ice models in place of the current physics-based melt ponds parametrisations.

We make use of data products which comprise some of the longest set of melt ponds observations. We show that it is possible to construct accurate point-wise emulators of melt ponds potentially suitable for column models such as Icepack (Hunke et al., 2021). Although neglecting horizontal processes, this choice avoids the needs for complex meshes and further grid requirements. In particular, we show that it is possible to predict to a reasonable degree melt pond fraction (MPF) on a dataset of 12.5 km × 12.5 km resolution from climatological input. Our work helps to demonstrate the viability of ML in
an area where parametrisations are necessary, typically bring considerably uncertainty, and where accurate simulations are of value to understand the future fate of sea ice.

    The paper is organised as follows: Section 2 introduces the datasets we use for the study, including basic analysis and data preparation. Section 3 describes the ML training and presents our findings from this. Section 4 gives our conclusions and discusses future work.

## 85  2   Datasets preparation and exploratory data analysis

### 2.1   Melt pond fraction target data

We aim at predicting MPF from given climate and sea ice variables. To achieve this, we leverage the "version 1.5" datasets from Istomina et al. (2020a, b) updating on Istomina et al. (2015a, b). The available datasets consist of daily values of the MPF over





the Arctic Ocean from both the Medium Resolution Imaging Spectrometer (MERIS) and Ocean and Land Colour Instrument

(OLCI) instruments onboard Envisat satellite - OLCI was partly developed to provide continuity with MERIS measurements. Together these provide daily data over the melt season (May to September inclusive) of MPF, spectral and broadband albedo for the periods of 2002-2011 (MERIS) and 2017-2022 (OLCI). This target data is gridded on a 12.5 km polar stereographic grid. Other valuable data exist, notably those from SHEBA (Perovich et al., 1999) and MOSAiC (Nicolaus et al., 2022) campaigns, yet they are both much shorter and localized around the ice camp thus lacking representativeness. Still in the past decades,

SHEBA has been the only observations available for testing melt pond parameterisations. The dataset we use is one of the most extensive one on MPF, an aspect that make it suitable for building parametrisations from deep learning. In the following, by MPF we intend the specific MERIS and OLCI products.

Figure 1 shows the timeseries of MPF as averaged over the entire Arctic (i.e. Northern Hemisphere) with a shading of 1 standard deviation either side of the mean. As expected, the data reveals that MPF has a qualitative seasonal cycle: starting

with low MPF over the whole Northern Hemisphere during the onset of the melt season (approximately May), MPF then rises to a maximum over the course of the summer, and decreases during refreezing as the Arctic approaches Autumn once more. The MERIS and OLCI periods are obviously different, both in the amplitude of the seasonal cycle and in the temporal variability. There is a priori no single explanation for these differences because the technological changes between the two satellite processing chains happen concurrently to a historical regime shift in Arctic sea ice properties (Sumata et al., 2023).

It is not decisive if the differences in the maximum and minimum values in the MPF between the 2002-2011 and 2017-2022 periods are attributable to differences in the MERIS and OLCI observational products or if there are differences for climatological reasons for instance, but our results favour the explanation of a change in the climate regime over that of sensors (we will elaborate more about this in Sect. 3.2).

Whilst we do not wish to predict a single global value, but the value at each individual geographical location, Fig. 1 nonethe-

less helps us understand the broad pattern of melt pond behaviour in the Arctic. Another way to explore the broad features of the data is with geographical maps. The dataset provides MPF from the 1st of May to the 30th of September for every year. Accordingly, here we can see each year beginning with low amounts of melt ponds in early Spring, rising as melt season onsets before refreezing begins in early Autumn. Time averaged geographical maps of our target data are shown in Fig. 2. These figure displays the time averages at each point for every possible instance in the windows of May-June, July and August-September.

i.e. removing all days without sea ice or with obstructed observations (by clouds for instance).

The plots show a similar pattern as is seen from the timeseries plots of Fig. 1: MPF starts off low uniformly across the Arctic, by July melting can be seen to onset most around the edges of the Arctic, around the Marginal ice zone (MIZ) more than central Arctic. Nonetheless melting does increase in these regions to a smaller extent. By August and September the central Arctic can be seen to return to lesser MPF values, with MPF higher at the edges of Russia and Canada. Large areas of no MPF values

both East and West of Greenland could indicate substantial ice retreat, as well as the scarcity of data in this MIZ region.



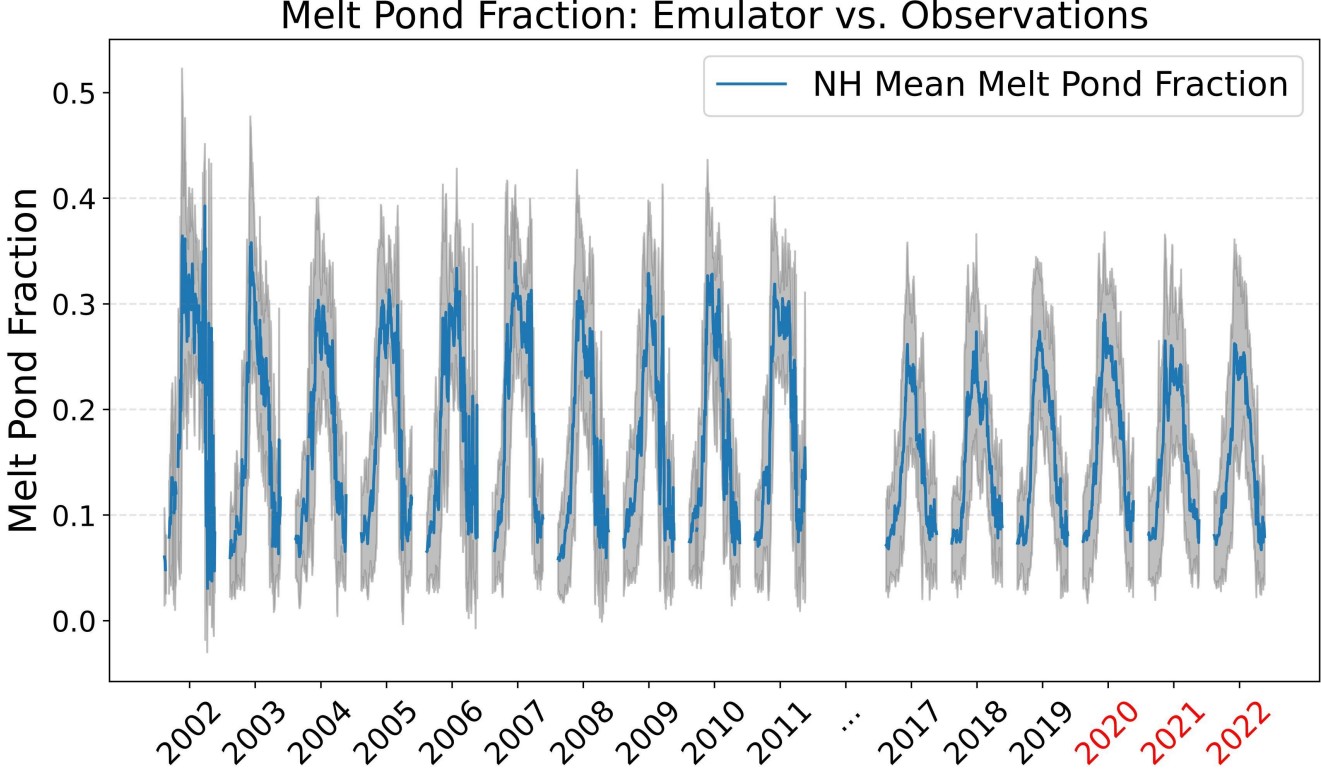

**Figure 1.** The target variable: MPF averaged across the NH. Shading represents 1 standard deviation either side of the average. Note that the dataset contains no negative MPF values; occasional negative values are due to calculating 1 standard deviation either side of the MPF values. The last three years - denoted in red - are used as test data.

## 2.2 Input data

We aim at predicting the MPF given knowledge of climate and environmental variables to which we have access. Therefore for features we take data from ERA5 hourly reanalysis dataset on single levels (Hersbach et al., 2023) at a resolution of $0.25°$ $\times 0.25°$, as well as sea ice age from the National Snow and Ice Data Center (NSIDC) and DMIOI-L4 data for sea ice fraction and ocean and ice surface temperature data (for DMIOI-L4 data: this study has been conducted using E.U. Copernicus Marine Service Information; doi.org/10.48670/moi-00123). A description of the variables used and data availability/duration is given in Tab. 1.

## 2.3 Data preparation

We build 0-dimensional emulators of MPF describing their evolution at a given location. To this end, we regrid the feature data to be on the same polar stereographically grid of the target MPF. For training data we take the years of 2002-2011 and




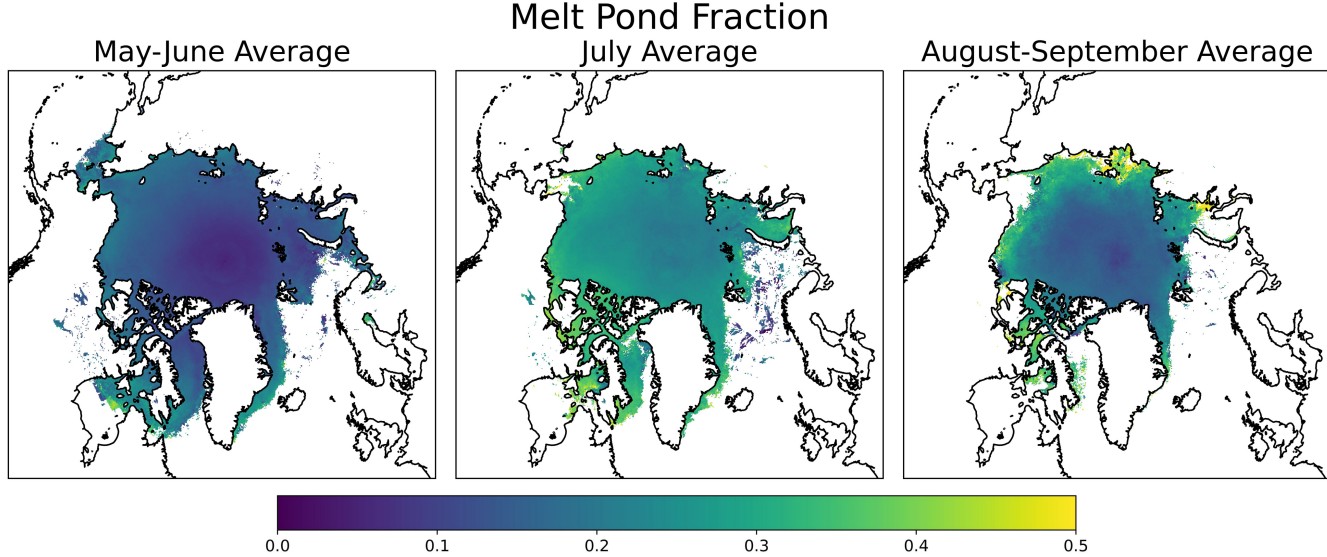

**Figure 2.** Plots of May-June, July and August-September averages of MPF - note that if even one observation was made between 2002-2022 for that point, that point is included. Thus whilst missing data is frequent, over the whole span much of the field can be seen. However, averages are calculated with different numbers of data contributing at each geographical location.

| Type | Variables | Source | Temporal Coverage | Spatial Coverage |
|---|---|---|---|---|
| Feature | 10 metre wind magnitude (U and V-wind) | ERA5 | 1948-present (hourly) | Global ($0.25° \times 0.25°$) |
| Feature | 2 metre dewpoint temperature | ERA5 | 1948-present (hourly) | Global ($0.25° \times 0.25°$) |
| Feature | Total Precipitation | ERA5 | 1948-present (hourly) | Global ($0.25° \times 0.25°$) |
| Feature | Surface Solar Radiation Downwards | ERA5 | 1948-present (hourly) | Global ($0.25° \times 0.25°$) |
| Feature | Surface Thermal Radiation Downwards | ERA5 | 1948-present (hourly) | Global ($0.25° \times 0.25°$) |
| Feature | Sea Ice Fraction | DMIOI-L4 | 1 Jan 1982 to 31 May 2021 (daily) | Lat $58°$ to $90°$, Lon $-180°$ to $180°$ ($0.05° \times 0.05°$) |
| Feature | Analysed ST | DMIOI-L4 | 1 Jan 1982 to 31 May 2021 (daily) | Lat $58°$ to $90°$, Lon $-180°$ to $180°$ ($0.05° \times 0.05°$) |
| Feature | Age of Sea Ice | NSIDC | 1 Jan 1984 to 31 Dec 2022 (daily) | Lat $29.7°D$ to $90°N$, Lon $-180°$ to $180°$ ($12.5km \times 12.5km$) |
| Target | Melt Pond Fraction | MERIS/OLCI | 01/05/2002-30/09/2011[1] & 01/05/2017-30/09/2022[2] | Polar Stereographic ($12.5$ km $\times$ $12.5$ km) |

**Table 1.** Summary of Feature and Target Data. All data used are described in terms of whether they are a feature or a target, the source of that data, as well as the temporal and spatial coverage. These are then regridded to the polar stereographic grid as shown in Fig. 3.




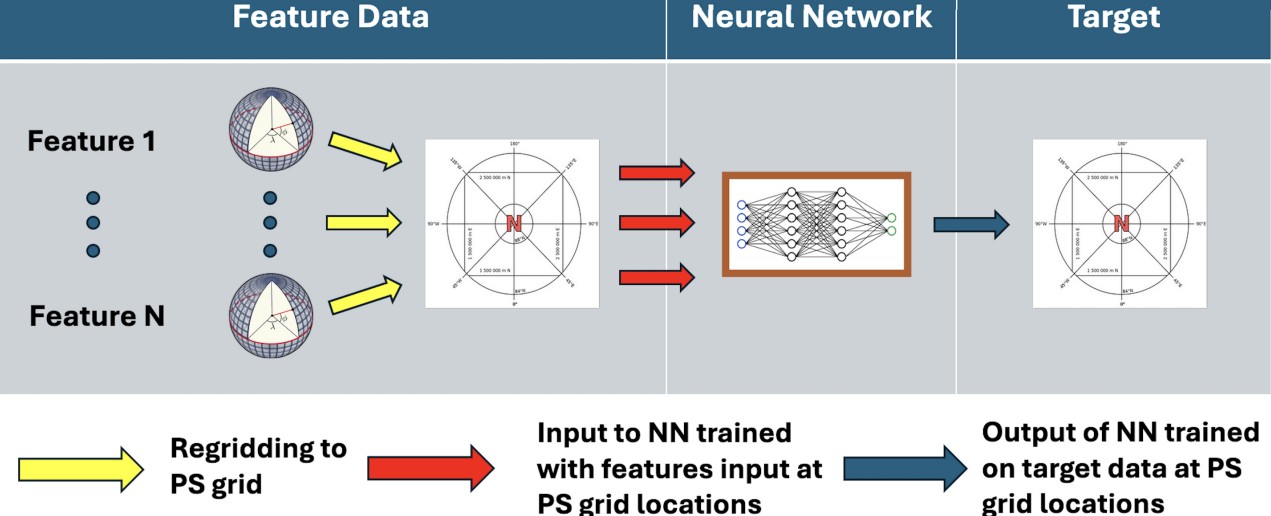

**Figure 3.** Schematic representing data preparation and training workflow. First all feature data from different sources is regridded from latitude-longitude grids, to be on the same polar stereographic grid as the melt pond observational data. At each point then on the polar stereographic grid input data can be fed to the neural network, where the output is compared to the observations at this exact location. The neural network is blind to location and training occurs in the same way at every location.

2017-2019 (13 years) and as test data the period 2020-2022 (3 years). The 12.5 km×12.5 km grid represents a dataset of daily values at 896×608 locations for melt seasons over 16 years.

Clouds can impair observations of MPF and albedo, thus not all geographic points are observable over the Arctic at any given time. Furthermore, some days of observations are missing for all locations. Therefore, as with many observational datasets, this dataset represents an "incomplete" dataset (unlike "perfect" or "model generated data"). Rather than use gap filling techniques (where then our ML algorithm would be learning in part this gap filled data) individual daily points with missing values are removed from the dataset. Points for which there is data available are incorporated into our analysis and the data are summarised in Tab.1 Figure 3 illustrates schematically the workflow we adopted to construct the MPF emulator.

## 3 Building an emulator of MPF from observations

### 3.1 Feature importance

With the broad characteristics of our dataset described, we use mutual information to assess the importance of our features on the target data. Feature selection can help identify which inputs have a larger impact on the output, and can help identify the importance of each variable and potential physical mechanisms at play. Mutual information (Shannon, 1948) measures the





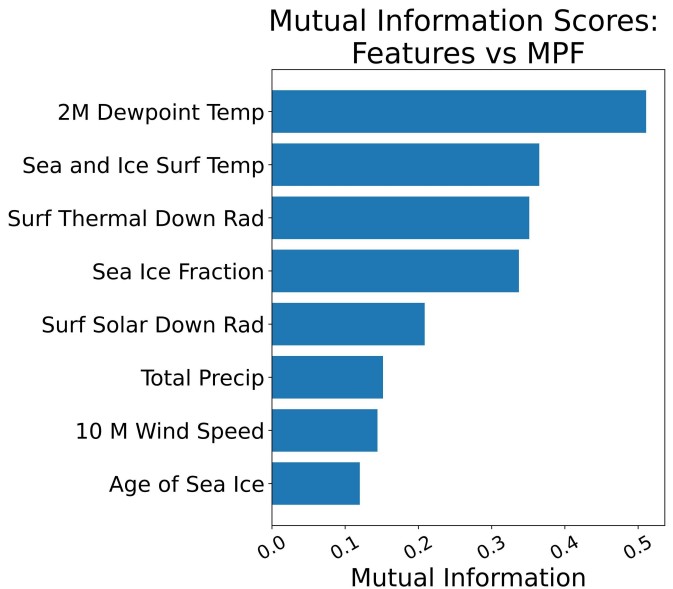

**Figure 4.** Mutual information between features (along the y-axis) and the MPF target.

amount of information that one, or a group of random, variable contains about another variable. It measures the dependency existing between two groups of continuous or discrete random variables.

With $X$ and $Y$ both being random variables with probability density functions given as $P_X$ and $P_Y$ respectively, and with domains given by $\mathcal{X}$ and $\mathcal{Y}$, their joint probability density function is defined as $P_{(X,Y)}$. Then the *mutual information* between $X$ and $Y$ is defined by (e.g., Frénay et al., 2013):

$$I(X;Y) = \sum_{x \in \mathcal{X}} \sum_{y \in \mathcal{Y}} P_{(X,Y)}(x,y) \ln \left( \frac{P_{(X,Y)}(x,y)}{P_X(x)P_Y(y)} \right) \mathrm{d}x\mathrm{d}y. \tag{1}$$

Mutual information is one way to perform a feature selection. Others exist and can offer different interpretations and nuances, yet mutual information has an advantage of being able to detect non-linear relationships between features and targets. In our precursor study, Driscoll et al. (2024), we used mutual information to rank the features' importance for the level-ice melt pond parametrisation and it was shown that rather than 15 variables used to predict melt pond properties as given by the physical level-ice scheme, pond area and height could be predicted relatively well even with 3 out of the 15 variables. Surface and air temperatures were reported to be of slightly more importance than top melt rates or the area of level ice (see Fig. 10 in Driscoll et al., 2024). This support our choice to use fewer variables in this current study - those which we deemed potentially key (see Table 1). To include some information about topography we included age of sea ice as a feature - older sea ice is generally more ridged, whilst younger sea ice is flatter and allows ponds to spread more widely.

Figure 4 shows the mutual information scores for the target variable with respect to the selected features. "Thermodynamic" variables, *i.e.* 2 m dewpoint temperature, sea and ice surface temperature, downwards surface thermal radiation and sea ice



fraction, appear to be the most influential on the MPF, with downwards solar radiation at the surface playing an intermediate role. Other data such as total precipitation, age of sea ice and wind speed (which plays a role in "advection") have secondary roles when judged by mutual information scores. This is consistent with the primary drivers being the temperature/energy available to cause melting. As seen in Driscoll et al. (2024), precipitation rates play a smaller role than these. We see also that

the wind speed, which mainly control the sea ice drift, plays a lesser role too. This evidence further corroborates our strategy of developing a column emulator, where horizontal-like interactions are assumed to be of secondary order at this spatial resolution. We recall that the first motivation to create column emulators was the same spirit as stated for the Icepack column model: by creating a column thermodynamic model it makes possible to incorporate it in any widersea ice model or GCM irrespective of complex grids or meshes. It is much easier to embed a column thermodynamic model than a 3D thermodynamic sea ice model.

## 170  3.2  Observationally trained emulator

With the feature analysis of the previous section in hands, we chose to explore the multi-perceptron model. Hyperparameter optimisation is performed as in Driscoll et al. (2024) using the Hyperband algorithm (Li et al., 2018), a novel bandit-based approach (which is a form of random search), where we searched over a hyperparameter space consisting of the number of nodes, layers, and types of activation functions (our loss function was fixed as MSE). We used the one year (2019) before the

test data period (2020-2022) as validation data. After the automatic optimisation we have also further manually explored some parameters. We found out that a smaller network of fewer nodes and layers, and thus faster to train, gave indistinguishable performance as compared to the deeper networks chosen by the automatic hyperparameter tuning. The major advantage of hyperparameter optimisation was in informing us on the choice of activation functions.

The procedure has chosen a deep neural network consisting of 10 hidden layers of 10 activation nodes each, using the ReLU

activation function, with a final output layer of a single node using the sigmoid activation function (as we predict a fractional value bound by 0 and 1). The number of epochs is set to 30 during training (sufficient to allow test loss to plateuax) and we made use of the Adam optimiser Following this, we assessed the performance of the emulator on the test MPF data.

The timeseries data of the results are shown in Fig. 5, together with the NN predicted MPF and the test target data. From this figure we can see that the emulator shows a very strong similarity to the observed MPF, with the emulator able to predict

very well the seasonal behaviour of the low MPF, including the rise and then later decline of MPF during late Summer and Autumn refreezing over the Arctic. The emulator is able to capture more nuanced events, such as a large scale refreezing as seen in e.g., June/July 2021. Although we do not investigate in detail the precise cause of this fast refreezing, the fact that the emulator captures this event specific to this year in the test data indicates that the emulator not only captures a broad seasonal behaviour but can capture unseen physical events in the Arctic.

We include a climatology as a baseline for comparison with our emulator. The climatology is created by taking the average value for each day and each grid point from the training data and using this to predict the MPF for that given day on the test data.

There is a strong seasonal dependence seen in melt pond evolution: for instance the Arctic is frozen over at the start of Spring, then undergoes melting during parts of the Spring and Summer, before refreezing. Therefore the climatology is potentially a





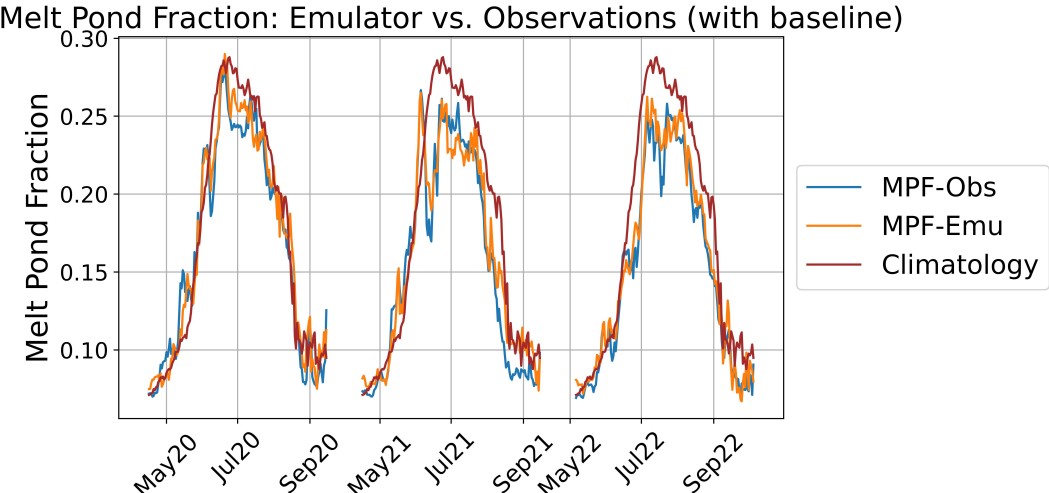

**Figure 5.** NH averages of observed MPF (orange) vs. predicted MPF given by the emulator (blue) and a climatology from the training data as a baseline.

reasonable predictor in the Arctic for MPF. The domination of solar influences, and periods of total darkness and light play a large role in the Arctic. Nonetheless our emulator outperforms this climatology quite substantially. Firstly, the climatology in general is worse than the emulator (over all regions our emulator's R2 score of 0.63 for MPF on the test data compares favourably to an R2 score of 0.31 for the climatology's prediction of MPF on the test data). Secondly, the climatology is incapable of responding to precise physical events, such as the short rapid drop and then increase of MPF during Spring/Summer in 2021 seen both in the emulator and observations.

The performance of our emulator across the Arctic is investigated in Fig. 6, where we show the spatial distribution of the R2 scores. In constructing these geographically-dependent R2 scores, we take all the points available on the test data at that given location. After analysis, we split this into two images. In Fig. 6a we show the regions where R2 scores are between 0 and 1 (inclusive). In Fig. 6b we show the regions where R2 scores are negative. When constructing R2 scores it was noted that a handful of locations gave large negative R2 scores compared to the overwhelming number of locations where the emulator performs well.

Figure 6b shows where the emulator should not be used. Plotting these on the same scale as Fig. 6a would obscure Fig. 6a from view. We note that the data corresponding to the regions of worse performance are close to the ice edge - where training data is poorer due to the absence of summer ice, and advection can no longer be neglected. A few occurrences of negative correlation are also visible in open water areas, which have escaped the satellite MPF quality checks. Nonetheless over most of the Arctic region we see that the emulator perform satisfactorily and in several instances captures extremely well the predicted MPF on unseen test data.

Our target dataset is composed of MPF derived from two different sources. To highlight this aspect, Fig. 7 shows the performance of the emulator on both the training and test set for both sources of data. Here we can see that the emulator, that



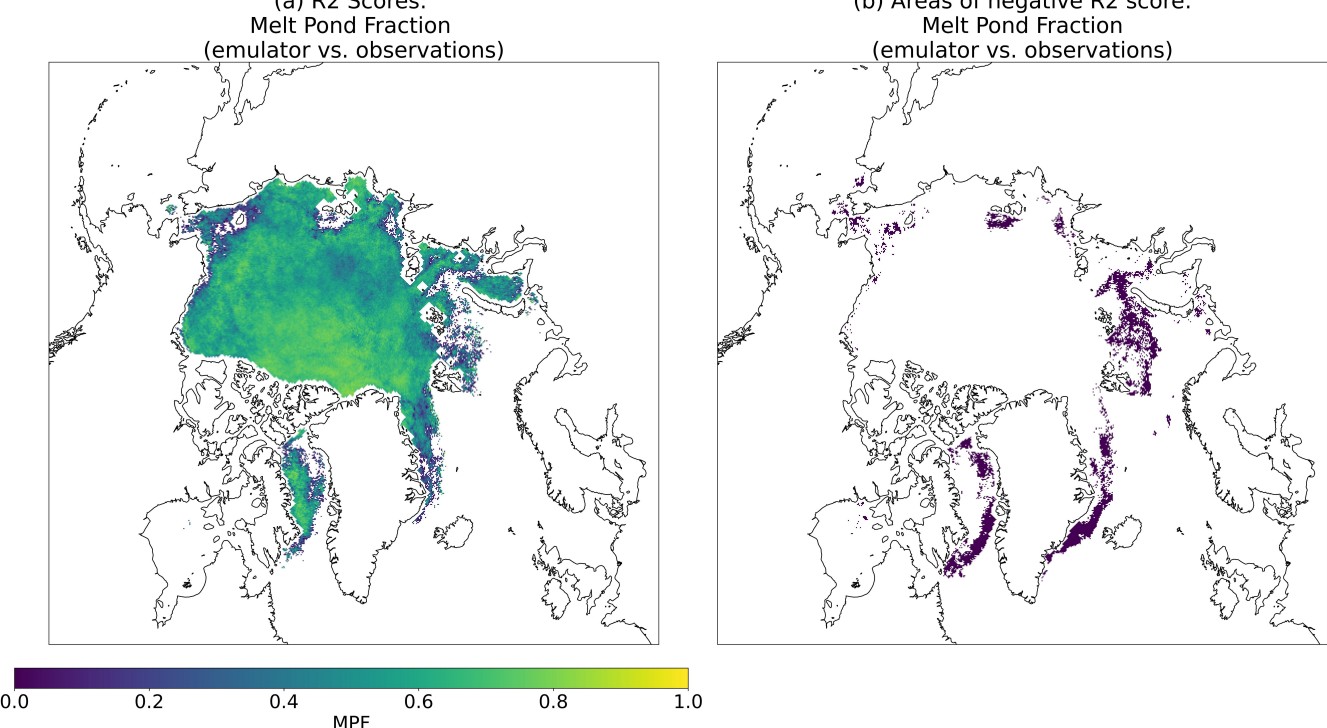

**Figure 6.** Geographically dependent R2 scores for the emulator on the test data (left) and areas where R2 scores are less than zero (right).

is trained on both the MERIS and OLCI data, is capable of learning well the magnitude and specific behaviours and events in both datasets. Most importantly, the shift between OLCI and MERIS is partially predicted by the emulator. Since our emulator relies only on environmental inputs that are independent of the OLCI and MERIS retrievals, this behavior suggests that the shift can be partially explained by physical trends that have a footprint in the input feature, and is not solely due to differences between the two satellite processing chains. By capturing events in both datasets, such as rapid brief refreezing seen in the years of 2007, 2008, 2009, and the physical phenomena as noted in the test dataset earlier, we can see that the emulator is able to adapt and learn to a reasonable degree the full observational record, and capture specific events.

Driscoll et al. (2024) showed that it is possible to emulate melt pond properties from model data. Using observational data, we have shown that a pointwise emulator that should be relatively easy to implement in climate models is thus able to predict to a reasonable degree MPF. In the conclusions and future work section we summarise our work, as well as reflect on the limitations and extensions of such an emulator.



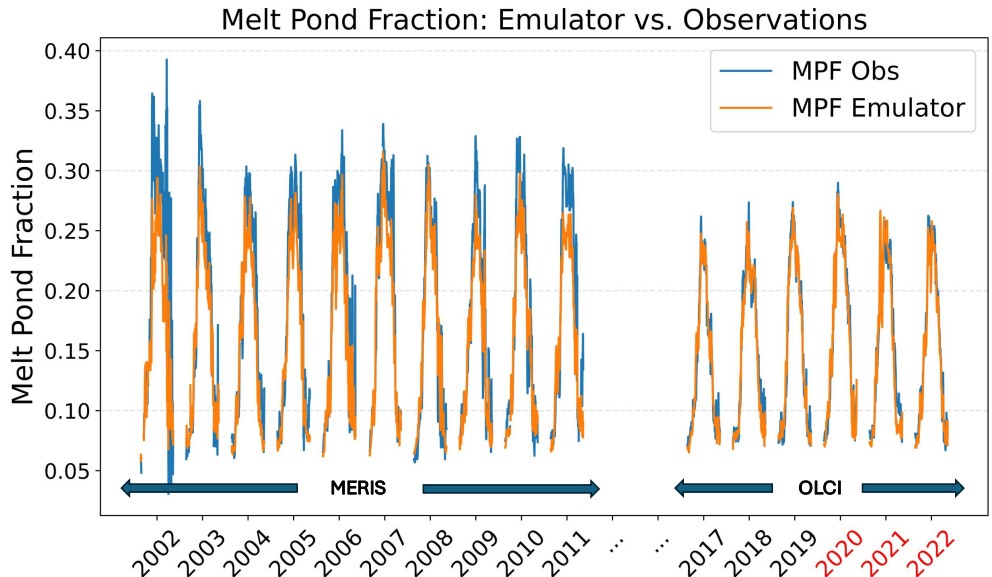

**Figure 7.** Emulator and MPF observations when compared over the whole training and test set. The training period 2002-2011 is based on MERIS and the second period 2017-2022 encompassing the training and test period is based on OLCI (the red years on the xticks indicate the test data).

## 4   Conclusions and future work

In this paper, we show that a pointwise emulator can successfully predict MPF using atmospheric and sea ice data at the same location. We anticipate that this emulator, once included in a full sea ice model, can avoid the timing errors that can affect presently available MPF parameterisations (Liu et al., 2015).

To achieve this aim, we considered variables important for sea ice and used in sea ice models, such as near surface and surface temperatures, wind, precipitation, and sea ice fraction. These were motivated not only by conventional wisdom, and expert opinion but also results from the findings of Driscoll et al. (2024) where an emulator of an existing state-of-the-art physical melt ponds parametrisation was derived. Data were collected from a variety of sources, then processed and regridded so that all features and targets shared the same polar stereographic grid. This pipeline is shown in Fig. 3.

We trained neural networks using this data to emulate the observed response. The emulator performs well in predicting MPF from observational variables. It outperforms a climatology and captures specific events that the climatology does not - indicating that the emulator is responsive to specific weather events.

Whilst our emulator does not perfectly match the melt pond state everywhere - performance varies regionally - the fact that the emulator performs well over most regions indicates at least a substantial part of the process of melt pond formation

shares some core and common physics across geographical areas. A notably different, i.e. worse performance was found in the marginal ice zone (MIZ) - where neglecting advection is limiting and data quality is substantially lower. Under future climate





change, Arctic sea ice will increasingly resemble MIZ conditions (becoming thinner and more fragmented). Our work also highlights the critical importance for both better observations and understanding in the MIZ. Nonetheless our emulator has a very good overall performance (e.g., Fig. 5).

Driscoll et al. (2024) was the first study to show it is possible to emulate melt pond processes on sea ice from model data. In the current study we move a step ahead and show that it is possible to successfully predict MPF from observational data. This stream of research is paving the way for the development of other and possibly more sophisticated data-driven emulators of melt ponds - removing the need for parametrisations based on simplified geometry and physics, and contributing furthermore to the wider discussion of using data-driven methods to replace traditional parametrisation approaches.

There are potential caveats in our study: we show that it is possible for a neural network to learn with good success the MPF product - our emulator relies on the accuracy of the satellite product considered as truth. The MPF data used here is itself a product, relying on the melt pond detector algorithm of Zege et al. (2015). As a result, it may be necessary to update our emulator to accommodate the successive improvements in the MPF product used as the target. Whilst datasets and algorithms will continue to be updated, these algorithms are not free of assumptions relating to sea ice and melt ponds.

A key part of the data-driven methods is their non-parametric nature. One possible way to create a emulator without relying on physical models or underlying parameter assumptions would be to construct a melt pond emulator entirely from detection of individual melt ponds from high-resolution images - albeit this is potentially much more labour-intensive. Exploiting algorithms on the large amounts of observational melt pond data, the pipeline presented here could be extended to construct a fully data-driven emulator as part of future work.

Istomina et al. (2020a) pointed to the lack of melt ponds being incorporated in any climate model. Diamond et al. (2024) performed simulations of HadGEM3.1 incorporating melt ponds to explore the role of inclusion of melt ponds as regards their effects on climate sensitivity. Our study provides an emulator that provides a candidate for use in climate modelling. Extrapolating the MPF emulator from the observed period 2002-2022 to different climates requires a leap of faith, but the emulator still represents a more representative estimator than empirical parametrisations that were calibrated on a single ice camp. In designing our emulator we specifically designed a pointwise emulator (1D) for easy incorporation into sea ice and climate models. Future work could involve coupling this emulator into the neXtSIM-DG sea ice model (Rampal et al., 2019; Richter et al., 2023) and assessing the performance of such a model in comparison to the topographic melt pond scheme (Flocco et al., 2010) which is currently implemented in neXtSIM-DG. It is not uncommon in climate modeling that an accurate 1D column parametrisation is made inaccurate by the numerical implementation of the full 3D model (for example vertical mixing models are notoriously performing poorer when used in 3D ocean models). However in the case of MPFs the only numerical schemes at play are related to horizontal advection which may incur some degree of smoothing depending on the choice made in each sea ice model. Still, the numerical diffusion of sea ice is rather slow compared to the large spatial scales of appearance and refreezing of melt ponds, so we only expect a minor smoothing effect due to horizontal sea ice numerics.



*Code and Data availability*

Code for the analysis and training used in this study is available from the authors upon request. The data used in this study are available directly from the operational centres as described in Table 1 and as thanked in the acknowledgements.

*Supplement*

None

*Author contributions*

Conceptualization: SD, AC, JB, LB, EOO, MB. Analysis and Visualization: SD. Interpretation of results: SD, AC, JB, LB, EOO, MB, ASL. Writing (original draft): SD. Writing (reviewing and editing original draft): SD, AC, JB, LB, EOO, MB, ASL.

*Competing interests*

The authors declare that they have no conflict of interest.

*Acknowledgements*

The authors acknowledge the support of the project SASIP funded by Schmidt Sciences (Grant number G-24-66154). Schmidt Sciences is a philanthropic initiative that seeks to improve societal outcomes through the development of emerging science and technologies. SD thanks the NERC Earth Observation Data Acquisition and Analysis Service (NEODAAS) for access to compute resources for this study. SD would like to thank Daniel Clewley and David Moffat for assistance with using 290 NEODAAS's Massive GPU Cluster for Earth Observation platform. CEREA is a member of Institut Pierre-Simon Laplace. SD would like to thank Larysa Istomina for her discussions and making the Bremen Sea Ice data available. Regridding of the data was done using the Climate Data Operators (CDO) regridding tools (Schulzweida, 2022). Support for the melt pond fraction dataset used here comes through EU project SPICES and DFG project REASSESS, DFG SPP 1158, grant number 424326801. AL was supported in part by the NERC National Centre for Earth Observation. The authors declare no competing interests.



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
