# Peer review of "Data-driven emulation of melt ponds on Arctic sea ice"

_EGUsphere, 2024_

## Referee Comment (RC1)

Review of "Data-driven emulation of melt ponds on Arctic sea ice" by Simon Driscoll1 et al.

**Summary:**

The positive feedback effect of melt ponds accelerates the Arctic sea ice melting process, significantly impacting the sea ice mass balance. However, statistically based physical parameterization schemes still exhibit significant uncertainty in representing subgrid-scale melt pond evolutions. In this study, the authors trained a machine learning emulator based on satellite observations as an alternative to the melt pond parameterization in a column sea ice thermodynamic model (i.e., Icepack). They emphasized that this emulator has the potential to be further integrated into climate models.

Overall, this is timely work, as the melt pond fraction on Arctic sea ice is increasing, and accurately simulating melt ponds can reduce the uncertainty of future climate projections in climate models. Regrettably, however, I find the current manuscript is not as well prepared as it should be for submission; the text makes for uncomfortable reading. Especially since the method is poorly introduced, does not convince the importance of its use to conclude, and does not meet the quality and reputation of The Cryosphere, I have to recommend **rejecting** its publication. Please find my specific comments below.

**Specific comments:**

1. Methods

The description of the method is so confusing that I cannot discern how the authors trained the emulator. At least the following points hinder my understanding:

(1) The dataset splitting

The authors said that the training dataset is that the 2002-2011 MERIS data and the 2017-2019 OLCI data (lines 130 and 132), but the data in 2019 were also used in the model validation (line 174)?

(2) The feature data

As listed in Table 1, the DMIOI-L4 sea ice fraction and analysed ST (end on 31 May 2021) are the "features". If I understand correctly, it means they are the "inputs" for the emulator. Therefore, during training and testing, the emulator's inputs should be consistent. However, data for these two variables is only available up to May 31, 2021. Does this mean that after this date, up to 2022, the emulator did not use these two variables as inputs? Or were data from another dataset used instead?

(3) Training of the emulator

I commend the authors for using the Hyperband algorithm to automatically optimize the model's hyperparameters and obtain the best combination. However, I am confused as to why manual hyperparameter tuning was also performed. According to the description below (line 179), the authors ultimately employed a fully connected neural network with 10 hidden layers and 10 nodes per layer. This configuration seems to have been determined by the Hyperband algorithm rather than manual tuning. If the intention is to discuss the impact of different hyperparameters on the results, I

suggest moving this content to the discussion section.

(4) Applicability of the emulator

What surprises me is that the authors, for each grid point, fit the daily MPF using features such as daily 10-meter wind speed and other variables. This emulator, which can be understood as a simple "multivariate nonlinear regression" model, even if trained to be very realistic, cannot be proven as a substitute for parameterization schemes. The input-output scenarios of this emulator do not align with those in model parameterization schemes, which typically operate with shorter time steps. Therefore, I am not convinced the authors have provided sufficient evidence that this emulator is capable of replacing the model's parameterization schemes.

(5) Significance of mutual analysis

I did not fully understand why the authors calculated the mutual information scores for each feature in this section. All features were ultimately used to establish the emulator, even though some of them might be less important, right? This part of the analysis seems more appropriate for moving into the discussion section to enhance the interpretability of the emulator.

2.  Data

I have strong concerns about the "version 1.5" MERIS and OLCI data shown here. I did not check the v1.5 dataset, but its updated version (Istomina et al., 2023) revealed an overall positive trend (+0.15% to +3% per decade) of the Arctic MPF (also can be seen in their Figures 9 and 12). I'm unsure if I missed something, but there indeed are some other observations that support the increasing Arctic MPF (e.g., Feng et al., 2022; Xiong and Ren, 2023), contrasting with Figure 1 in this manuscript. I strongly suggest that the authors check for errors in the way they handle the data.

3.  Results

Section 3 presents results in a structurally disorganized manner. A more coherent approach would be to present the model validation results first, followed by the test results.

The current presented results have at least two critical shortcomings: the performance and validation of the emulator. The authors described "the emulator shows a very strong similarity to the observed MPF" in line 184, however, from my perspective, although the emulator shows overall similarity with observations in the test set, there are many obvious mismatches (see areas circled in green in the figure below). Thus, the authors' description of the results is imprecise and overly colloquial. On the other hand, I believe the current validation approach is ineffective. It is necessary to compare the emulator with the original parameterization scheme on the same test set, rather than only comparing it against the climatology (lines 190-192). In other words, I believe the authors have not achieved their stated goal of replacing the physical parameterization scheme with the emulator (also refer to my fourth comment on the methods).

Furthermore, I suggest that the authors validate the emulator's effectiveness using MOSAiC data (e.g., Webster et al., 2022) and compare it with the original

parameterization scheme, which would render the study more comprehensive.

[Figure]

Other issues:

Please note that because there are numerous language and formatting issues in this manuscript, only several of them are listed below. To improve the quality of your manuscript, I recommend thoroughly revising the language to ensure a smoother flow and clarity.

- The language of this paper is excessively verbose and lacks academic rigor. I mean, one should not use vague terms such as "very" to describe results (e.g., line 60, line 62, line 66, line 184, line 244).
- Figure 1: Which line represents the "emulator" mentioned in the title?
- Figure 3: I do not think this simple training workflow worth a schematic figure to illustrate. The only information I can get from this schematic figure is that the authors interpolated the features onto a widely used polar stereographic projection grid.
- Lines 187-189: I am unsure if the emulator has not seen this "large scale refreezing" in the test dataset, and thus, I am not convinced by this statement.
- Lines 228-229: What does this sentence mean? Not clear.
- The formatting of the references is highly messy.

References:

Feng, J., Zhang, Y., Cheng, Q., and Tsou, J. Y.: Pan-Arctic melt pond fraction trend, variability, and contribution to sea ice changes, Global and Planetary Change, 217, 103932, https://doi.org/10.1016/j.gloplacha.2022.103932, 2022.

Istomina, L., Niehaus, H., and Spreen, G.: Updated Arctic melt pond fraction dataset and trends 2002–2023 using ENVISAT and Sentinel-3 remote sensing data, https://doi.org/10.5194/tc-2023-142, 22 September 2023.

Webster, M. A., Holland, M., Wright, N. C., Hendricks, S., Hutter, N., Itkin, P., Light, B., Linhardt, F., Perovich, D. K., Raphael, I. A., Smith, M. M., Von Albedyll, L., and Zhang, J.: Spatiotemporal evolution of melt ponds on Arctic sea ice,

Elementa: Science of the Anthropocene, 10, 000072, https://doi.org/10.1525/elementa.2021.000072, 2022.

Xiong, C. and Ren, Y.: Arctic sea ice melt pond fraction in 2000–2021 derived by dynamic pixel spectral unmixing of MODIS images, ISPRS Journal of Photogrammetry and Remote Sensing, 197, 181–198, https://doi.org/10.1016/j.isprsjprs.2023.01.023, 2023.

---

## Referee Comment (RC2)

Peer review of "Data-driven emulation of melt ponds on Arctic sea ice" by Driscoll et al.

The manuscript presents a Machine-Learning (ML) model intended for estimating melt-pond fraction on sea ice based on meteorological variables and developed and tested against spectrometer satellite data of Melt-Pond Fraction (MPF). The work appears solid but the manuscript is somewhat short on methodology and review of earlier work. In addition, the manuscript seems to claim that melt ponds have not been taken into account in climate models before, which is not the case. Here follows some mostly minor comments and suggestions that the authors can take into account before acceptance for publication is considered.

Comments and suggestions

1. In particular in the abstract line 5, the manuscript seems to claim that melt ponds are not included in GCMs, and that melt ponds therefore play no role in IPCC future projections of sea ice. This appears not to be the case, indeed many models have included parameterisations of melt ponds, see e.g Sterlin et al, Ocean Modelling, 2021, and references therein, as well as Roeckner et al., Journal of Advances in Modeling Earth System, 2012, and Diamond et al, Journal of Climate, 2023.

In addition to correct this aspects of historical uses of melt-pond parameterisations in ECMs, the manuscript should also provide a much more extensive review of historical efforts to provide melt-pond parameterisations, including for instance reviews of Lühtje et al., J. Geophys. Res – Oceans, 2006, and Flocco and Feltham, J. Geophys Res. - Oceans, 2007.

2. Test against SHEBA. In order to test performance of the developed ML model with other types of data and to results from other parameterisations (e.g. Holland et al, J. Clim., 2012), the SHEBA data may be applied.

3. Section 2.1 first paragraph: Expand the description of the MPF data, e.g. regarding how MPF is obtained from these data. Are there missing data and due to what?

4. L116-119: The conclusion regarding change in pattern between the two observational data set beibg due to instrumental differences or climate changes would be better explored if training data were chosen for only MERIS, 2002-2011.

5. First two paragraphs of section 3.2: Please more extensively describe the procedure of constructing the model, including finding the hyperparameters, in a way that none ML experts can follow. Shortly describe all "ML words". "Plateaux" → "converge".

6. L69-72: Please first describe the earlier studies before indicating differences to yours.

7. Fig. 1 caption: Explain how the standard deviation is calculated, what are the input data?

8. L124: Write out the abbreviation "DMIOI-L4".

9. Fig. 2 caption: In the first sentence before "note that" indicate the period averages are taken over. The "note that" part is quite awkwardly formulated, perhaps "Note that if at least one observation exists for a given grid point this point is included".

10. L135-137: The sentence is awkwardly formulated.

11. L155: What is "level ice"?

12. L189: "unseen" seems to be exaggerated given that The ML model build on observational input. Perhaps it suffice to conclude that the model can reproduce MPF associated with stochastic weather variability.

13. Fig. 5: Why not show root-mean square error (RMSE) of model results? Perhaps also show anomaly relative to climatology of observations, in order to compare (and hereby regard climatology as a persistence model).

14. L197: What is "R2 score"?

15. Fig. 6: In (b) the scale of the shading is not indicated.

16. L255-259: This paragraph is not so clear.

Typos:

L92: "data is" → "data are", data are plural. Same in Fig. 2 caption. And Line 235.
L96: Remove "one".
L130: "of'->"as".
L245: "it is" → "that it is".

---

## Author Comment (AC1)

**Containing Author Comments**

**Reviewer 2**

Peer review of "Data-driven emulation of melt ponds on Arctic sea ice" by Driscoll et al.

The manuscript presents a Machine-Learning (ML) model intended for estimating melt-pond fraction on sea ice based on meteorological variables and developed and tested against spectrometer satellite data of Melt-Pond Fraction (MPF). The work appears solid but the manuscript is somewhat short on methodology and review of earlier work. In addition, the manuscript seems to claim that melt ponds have not been taken into account in climate models before, which is not the case. Here follows some mostly minor comments and suggestions that the authors can take into account before acceptance for publication is considered.

We acknowledge the reviewer's comments and have been very grateful for them. We believe going through them and adding these details and insights help yield a substantially improved manuscript.

Comments and suggestions

1. In particular in the abstract line 5, the manuscript seems to claim that melt ponds are not included in GCMs, and that melt ponds therefore play no role in IPCC future projections of sea ice. This appears not to be the case, indeed many models have included parameterisations of melt ponds, see e.g Sterlin et al, Ocean Modelling, 2021, and references therein, as well as Roeckner et al., Journal of Advances in Modeling Earth System, 2012, and Diamond et al, Journal of Climate, 2023.

We thank the reviewer for this comment. We apologise for the confusion we have caused. We intended not to imply that melt ponds are not used in GCMs or projections in any way, but that historically it has been often the case that GCMs are run without them, and that thus key processes for sea ice prediction are likely being missed. (A similar conclusion is given in Diamond et al. 2023 itself who point out that only 11% of CMIP6 models have an explicit melt pond scheme).

Our message was intended in the spirit of Diamond et al. 2023, but not clearly communicated by us, which we have updated. We also included these and multiple other references demonstrating where indeed the inclusion of melt pond processes shows them to have an important role on sea ice prediction. We have hopefully given a useful account therefore of the literature provided here, as well as other references.

In addition to correct this aspects of historical uses of melt-pond parameterisations in ECMs, the manuscript should also provide a much more extensive review of historical efforts to provide melt- pond parameterisations, including for instance reviews of Lühtje et al., J. Geophys. Res – Oceans, 2006, and Flocco and Feltham, J. Geophys Res. - Oceans, 2007.

We have updated our manuscript to include a detailed history review of melt pond parametrisations and approaches. We thank the reviewer as it substantially improves our paper.

2. Test against SHEBA. In order to test performance of the developed ML model with other types of data and to results from other parameterisations (e.g. Holland et al, J. Clim., 2012), the SHEBA data may be applied.

We thank the reviewer for this comment. The SHEBA campaign has represented a valuable dataset in the sea ice literature. Our study was designed around a different objective: the spirit of our model is to inherently create an emulator that is created and tested against broad-scale, long-term, pan-Arctic observations. The MERIS and OLCI datasets therefore are ideally suited for training and evaluating a data-driven emulator intended to generalise across the Arctic domain.

In this work we chose to evaluate our emulator's performance as whole against pan-Arctic data, which we feel is a strength. Other data have merit, but they have spatial and temporal limitations (thus can be affected by processes specific to one location and conditions affecting one year). These do not necessarily provide a representative testbed for the generalisation we wish to achieve. By demonstrating one can emulate directly from observations we feel this is a key first step, and we have included many potential future avenues in the paper now. We have highlighted the reviewer's point and included our motivational reasons much more in our text.

3. Section 2.1 first paragraph: Expand the description of the MPF data, e.g. regarding how MPF is obtained from these data. Are there missing data and due to what?

We have added in details on the MPF data, how it is obtained, missing data and its sources accordingly.

4. L116-119: The conclusion regarding change in pattern between the two observational data set beibg due to instrumental differences or climate changes would be better explored if training data were chosen for only MERIS, 2002-2011.

We have explored this by using training data that covers the MERIS only period of 2002-2011. Indeed the R2 score drops from 0.63 to 0.54 and the MSE increases from 0.0030 to 0.0038 (an increase of approx 27%). So we see that a MERIS only trained emulator causes a greater separation between observations and emulator in the second satellite era. That there still exists a good predictability when trained on MERIS data only to predict the other is still a positive result, but this test reduces our ability to attribute shifts to climate change. This has shaped our narrative and also motivated the inclusion of results from an additional melt pond fraction dataset where we see very good emulation scores. We have updated the conclusion to reflect the nuance above. We thank the reviewer for the suggestion and this test that has enlightened the discussion.

5. First two paragraphs of section 3.2: Please more extensively describe the procedure of constructing the model, including finding the hyperparameters, in a way that none ML experts can follow. Shortly describe all "ML words". "Plateaux" → "converge".

We thank the reviewer for this. This has been now done.

6. L69-72: Please first describe the earlier studies before indicating differences to yours.

Our work is substantially different and we have clarified this now – for example Peng et al. 2022 use neural networks and statistical temporal filters to fill observational gaps, and effectively interpolate over missing or obscured melt pond data so as to ensure continuity. In essence their work reconstructs missing melt pond data, using a statistical and ML interpolation system. Ours seeks to learn the physical mapping from climate data to melt pond fraction. Thus our

work importantly avoids statistical filling, uses only direct observational data and instead seeks to build relationships to physical variables. We have added additional information to clarify this and therefore show how our study is distinct.

7. Fig. 1 caption: Explain how the standard deviation is calculated, what are the input data?

We have explained this in the caption.

8. L124: Write out the abbreviation "DMIOI-L4".

Done.

9. Fig. 2 caption: In the first sentence before "note that" indicate the period averages are taken over. The "note that" part is quite awkwardly formulated, perhaps "Note that if at least one observation exists for a given grid point this point is included".

Amended. Thank you.

10. L135-137: The sentence is awkwardly formulated.

Thank you for pointing this out. We have rewritten this for clarity and flow.

11. L155: What is "level ice"?

The level-ice means not ridged, or deformed ice, and the associated 'level-ice' melt pond scheme developed by Hunke et al. (2013) is designed to represent the core concept that meltwater collects predominantly on level-ice, not on ridged or deformed ice. We have added this explanation to our text.

12. L189: "unseen" seems to be exaggerated given that The ML model build on observational input. Perhaps it suffice to conclude that the model can reproduce MPF associated with stochastic weather variability.

Apologies for our confusing wording– we intended "unseen" to refer to the fact it is test data. We agree and have reworded it so it now refers to capturing physical events in the Arctic associated with stochastic weather variability (in the test data).

13. Fig. 5: Why not show root-mean square error (RMSE) of model results? Perhaps also show anomaly relative to climatology of observations, in order to compare (and hereby regard climatology as a persistence model).

We have amended the figure to have alongside it the values of RMSE for easy comparison and thank the reviewer.

14. L197: What is "R2 score"?

We have added in an explanation of R2 score.

15. Fig. 6: In (b) the scale of the shading is not indicated.

We have added information now.

16. L255-259: This paragraph is not so clear.

We believe we have clarified this now.

Typos:

L92: "data is" → "data are", data are plural. Same in Fig. 2 caption. And Line 235. L96: Remove "one".
L130: "of"->"as".
L245: "it is" → "that it is".

We have amended all typos, except for keeping L235 as a mass noun.

---

## Author Comment (AC2)

**Containing Author Comments**

*Reviewer 3*

Review comment for 'Data-driven emulation of melt ponds on Arctic sea ice' in *The Cryosphere*

In this study, Driscoll et al. present a point-wise emulator for Arctic melt ponds, developed using pan-Arctic satellite observations and reanalysis data. They demonstrate its ability to predict Melt-Pond Fraction (MPF), which shows promise for integration into global climate models (GCMs).

Given the increasing prevalence of melt ponds due to ongoing global warming, research on melt ponds in polar regions is becoming increasingly critical in the context of climate change. However, while the study provides valuable insights, there remains considerable room for improvement.

Therefore, I recommend that the manuscript undergoes major revisions, and specific comments are outlined below.

We are extremely grateful for the reviewer's comments and insights, and by incorporating them it has improved our manuscript substantially.

1. The introduction section would benefit from further elaboration, particularly with regard to the context and rationale behind the study. The following points, though not exhaustive, serve as examples:

(1) Lines 65-69: Driscoll et al. (2024) have already developed an emulator for melt ponds based on model data. If this existing emulator performs adequately, what is the justification for advancing to an emulator built solely on observational data? A clearer explanation of the advantages and motivation for this shift would strengthen the rationale for the current work.

Thank you for this point. Our previous emulator was trained on model-generated melt pond data and therefore inherited the structural assumptions and sensitivities of the underlying scheme. However, by demonstrating that such an emulator could run stably within a model, we showed the feasibility and

potential value of using emulators - particularly if one could link climate variables directly to observational data.

Yet transitioning from model data to observational data in emulators has often been a prohibitive step in climate research. In contrast, the current study does just that and therefore demonstrates the next step: it learns from observed melt pond fraction, aiming to reduce uncertainty rather than reproduce it. We have clarified this in the revised text. We have also highlighted the significance of a first step in making an observational emulator in the context of wider climate research.

(2) Lines 70-72: The distinction between the approach used by Peng et al (2022), which involves "interpolating melt pond observations across a temporal series," and the methodology employed in the current study, which involves "regridding satellite data of melt ponds," needs further clarification. Specifically, how do these two techniques differ in terms of their methodology, accuracy, and overall Applicability?

Our work is substantially different and we have clarified this now – for example Peng et al. 2022 use neural networks and statistical temporal filters to fill observational gaps, and effectively interpolate over missing or obscured melt pond data to ensure continuity. In essence their work reconstructs missing melt pond data, using a statistical and ML interpolation system. Ours seeks to learn the physical mapping from climate data to melt pond fraction. Thus our work importantly avoids statistical filling, uses only direct observational data and instead seeks to build relationships to physical variables. We have added additional information to clarify this and therefore show how our study is distinct.

2. The methodology section would benefit from additional explanations to better justify the importance of the chosen approach. The following points, though not exhaustive, are provided as examples:

(1) Lines 102-103: Given the significant differences in the amplitude of the seasonal cycle and temporal variability, how might these factors affect the consistency of the data between the training and test periods? Additionally, considering that Arctic sea ice is expected to increasingly resemble MIZ conditions (Line 242), would using segmented training data lead to improved

results and potentially alter conclusions, particularly with regard to the identification of feature importance?

The slight difference makes our goal harder, thus we believe our emulator is doing well. We have also accommodated the reviewer's insight here by testing on another version of this dataset where the OLCI peaks do not suffer from this discrepancy: our performance (with a second trained emulator) improves. Success across two datasets adds to the robustness of our approach and the validity of our pipeline. We have added the details of this work and our scientific findings into our paper. We thank the reviewer for spotting this and therefore enriching our paper through this work.

We acknowledge the reviewer's second insightful point. Whilst the goal of our paper is to identify useful observational variables and build a pointwise emulator from observed inputs, such that it is broad-scale, long-term and pan-Arctic, it is absolutely possible that one might be able to build a "MIZ only" emulator in the future. It is likely that with distinct physical processes in the MIZ, new features would be required to emulate this well – for example those that represent interconnectivity in ponds, surface roughness and floe topography given a younger, flatter, yet more broken surface. There are multiple substantial challenges. Given greater uncertainties around quality of different MIZ data one would test many different data sources, campaigns and imagery to capture desired effects required by potentially new and necessary variables from such datasets (perhaps developing tools to super-resolve existing data). We have in our conclusions noted other challenges that come with this. For example, there is no single definition of the MIZ and such a study could include and test many definitions. We are in consultation with groups that have access to proprietary datasets for such purposes and are considering a research proposal to similar effects.

We are excited by the reviewers comment and have incorporated the reviewer's point both discussing the impacts of feature analysis that a "MIZ only" focus might have (including those with new necessary variables), as well as discussing now in more detail MIZ emulators as a potential future research direction. Ideally our study serves as a key first step – one that has often eluded other areas of climate research in building observational emulators – and helps set up future investigations including that one.

(2) Line 138: The description of the specific training process is overly simplistic and fails to capture the nuances of the neural network architecture employed in this study. A more detailed explanation of the training procedure, including the rationale behind the choice of neural network model, would enhance the clarity and rigor of the methodology.

We have added more detail behind the choice of neural network model, and the training involved to enhance clarity and rigor. We thank the reviewer for this point.

3. The results section would benefit from additional discussion to emphasize the significance of this study and its contributions to the field (1) Line 156: Can the conclusions regarding feature importance drawn from the model data be directly applied to a study based solely on observational data? It would be valuable to discuss the potential limitations or caveats when transferring findings from model-based to observational datasets.

We thank the reviewer for this. Indeed we believe it to be a significant step and we have added additional discussion on such significance – such as the difficulties faced in other areas of creating observational emulators as compared to synthetic data without gaps and noise.

We appreciate the second point fully. And we have added in a greater discussion on such caveats.

(2) Lines 164-165: Considering that Arctic sea ice is becoming thinner and more fragmented (Line 242), advection is expected to play an increasingly significant role in the evolution of melt ponds. However, this study only considers the influence of wind speed, while neglecting the impact of wind direction. Could the omission of wind direction potentially affect the study's conclusions, particularly in light of the changing dynamics of Arctic sea ice?

We thank the reviewer for this point. We therefore tested the emulator with and without wind direction (i.e. not only the magnitude) and the results are essentially identical including in the MIZ. However there was an extremely small, almost negligible, improvement by using the magnitude. Therefore we kept magnitude, and our conclusions are not affected. We have now added these

details to inform and enrich our paper. We thank the reviewer for raising this point.

4. Several typographical errors and omissions need to be addressed: (1) Figure 1: The distinction between observational data and model results is not clearly indicated in the image. It would be helpful to specify how these two data sources can be differentiated visually.

We apologise for this – this is indeed a typo and has been updated and clarified.

(2) Lines 172-173: Further clarification is needed regarding the "bandit based approach." A more detailed explanation of this method would enhance the reader's understanding of its relevance and application in the current study.

Done.

(3) Line 182: The "optimiser Following" should be corrected to "optimizer. Following."

Corrected.

(4) Figure 5: The figure legend and the image itself do not align. Additionally, the year should be added to the x-axis to provide clear temporal context for the data.

We have updated the figure accordingly.

(5) Line 196: The definition of R2 should be provided for clarity, as some readers may not be familiar with the metric or its specific interpretation in this context.

We have provided a definition of R2 score.

---

## Author Comment (AC3)

**Containing Author Comments**

**Review 1**

Review of "Data-driven emulation of melt ponds on Arctic sea ice" by Simon Driscoll1 et al.

**Summary:**
The positive feedback effect of melt ponds accelerates the Arctic sea ice melting process, significantly impacting the sea ice mass balance. However, statistically based physical parameterization schemes still exhibit significant uncertainty in representing subgrid-scale melt pond evolutions. In this study, the authors trained a machine learning emulator based on satellite observations as an alternative to the melt pond parameterization in a column sea ice thermodynamic model (i.e., Icepack). They emphasized that this emulator has the potential to be further integrated into climate models.

Overall, this is timely work, as the melt pond fraction on Arctic sea ice is increasing, and accurately simulating melt ponds can reduce the uncertainty of future climate projections in climate models. Regrettably, however, I find the current manuscript is not as well prepared as it should be for submission; the text makes for uncomfortable reading. Especially since the method is poorly introduced, does not convince the importance of its use to conclude, and does not meet the quality and reputation of The Cryosphere, I have to recommend **rejecting** its publication. Please find my specific comments below.

We thank the reviewer for their insights. A large effort with our co-authors has been made to substantially improve the clarity of our text and remove confusion we caused. Aided by the reviewer's insights, we have implemented notable scientific changes. We believe all together these have made a substantially improved paper that addresses the points made. We hope to convince the reviewer with the following answers and are incredibly grateful for the reviewer's input that led to these improvements.

**Specific comments:**

1. Methods

The description of the method is so confusing that I cannot discern how the authors trained the emulator. At least the following points hinder my understanding:

(1) The dataset splitting
The authors said that the training dataset is that the 2002-2011 MERIS data and the 2017-2019 OLCI data (lines 130 and 132), but the data in 2019 were also used in the model validation (line 174)?

We thank the reviewer for highlighting this and apologise for the confusion we caused. In machine learning, when the datasets are short it is common practice to use all validation and training data for final training, whereas test data is kept separately. This is particularly useful for the cryosphere, where data is often sparse or missing. We also refer the reviewer to point 3, where we have much more clearly clarified the minimal role hyperparameter tuning played and have moved it to the conclusions section in our paper as per the suggestion by the reviewer. In this section we then fully clarify our approach.

(2) The feature data
As listed in Table 1, the DMIOI-L4 sea ice fraction and analysed ST (end on 31 May 2021) are the "features". If I understand correctly, it means they are the "inputs" for the emulator. Therefore, during training and testing, the emulator's inputs should be consistent. However, data for these two variables is only available up to May 31, 2021.

Does this mean that after this date, up to 2022, the emulator did not use these two variables as inputs? Or were data from another dataset used instead?

This is a copy paste error. When originally looking into datasets many years ago the DMI had datasets that ran up to this time, now seamlessly extended. We use the full same dataset that covers the whole period, and this has been updated in our table. We thank the reviewer for spotting this error for us.

(3) Training of the emulator
I commend the authors for using the Hyperband algorithm to automatically optimize the model's hyperparameters and obtain the best combination. However, I am confused as to why manual hyperparameter tuning was also performed. According to the description below (line 179), the authors ultimately

employed a fully connected neural network with 10 hidden layers and 10 nodes per layer. This configuration seems to have been determined by the Hyperband algorithm rather than manual tuning.

If the intention is to discuss the impact of different hyperparameters on the results, I suggest moving this content to the discussion section.

Thank you for this suggestion which has made us improve the description of our approach. We have not ultimately used automatic hyperparameter tuning in our choice of final model. We did this manually and have clarified our wording which was confusing - hyperparameter optimisation of complex models was ultimately not necessary as simpler smaller models were equally good. As well as clarifying what we did we have moved this portion of the text to the discussion section.

(4) Applicability of the emulator
What surprises me is that the authors, for each grid point, fit the daily MPF using features such as daily 10-meter wind speed and other variables. This emulator, which can be understood as a simple "multivariate nonlinear regression" model, even if trained to be very realistic, cannot be proven as a substitute for parameterization schemes. The input-output scenarios of this emulator do not align with those in model parameterization schemes, which typically operate with shorter time steps. Therefore, I am not convinced the authors have provided sufficient evidence that this emulator is capable of replacing the model's parameterization schemes.

We thank the reviewer for this comment. We have clarified that our objective is not to show that such an emulator can replace the model's parametrisation schemes. Nor do we aim to emulate the full physical complexity of the evolution and formation of melt ponds. Instead, we seek to identify useful observational variables and learn the relationship between observed inputs and melt pond fraction – which is understood here as a proxy for albedo on sea ice and thus is a key variable relevant to the Arctic's energy budget and is integral in models. By showing that a relationship exists between key observed variables, and particularly those often used in models, we believe this is an important step – as well as informative for the physical understanding of melt ponds.

We have discussed that showing this relationship exists is valuable in itself - going from emulators based on synthetic/model data to those based on noisy, sometimes missing, observational data has been a difficult, major and often prohibitive step in many areas of climate research. We expand on the significance of the research. The design choices that we made for simplification (no advection, daily averaged values) might be prohibitive for replacing a model parametrisation, and we now discuss this in more detail, but our study makes us optimistic about the future steps for data-driven developments in the field. Our manuscript title has been updated to highlight we are solely targeting melt pond fraction. We have added text to highlight the challenges that such emulators might have in replacing model parts, such as advection and time stepping issues, as well as the complex processes involved in melt pond formation, and thus in physical models. In doing so we also consider where data-driven methods might be useful in aiding future scientific inquiry and enriching model processes - such as in identifying shifts in fractal dimensions and interconnectivity in melt ponds from imagery.

(5) Significance of mutual analysis
I did not fully understand why the authors calculated the mutual information scores for each feature in this section. All features were ultimately used to establish the emulator, even though some of them might be less important, right? This part of the analysis seems more appropriate for moving into the discussion section to enhance the interpretability of the emulator.

We thank the reviewer for their useful comment, and your interpretation is correct. We have indeed moved the mutual information section to the conclusions to enhance the interpretability of the emulator.

2. Data

I have strong concerns about the "version 1.5" MERIS and OLCI data shown here. I did not check the v1.5 dataset, but its updated version (Istomina et al., 2023) revealed an overall positive trend (+0.15% to +3% per decade) of the Arctic MPF (also can be seen in their Figures 9 and 12). I'm unsure if I missed something, but there indeed are some other observations that support the increasing Arctic MPF (e.g., Feng et al., 2022; Xiong and Ren, 2023), contrasting with Figure 1 in this manuscript. I strongly suggest that the authors check for errors in the way they handle the data.

We thank the reviewer for their valuable insights on this as it has improved our paper. We can confirm the data handling to produce Figure 1 is indeed very straightforward (using built in python functions), and that the step change drop between the MERIS and OLCI is a real feature, but a limitation, of using the "V1.5" dataset. We thank the reviewer for pointing this out as we now have also tested our approach on the updated V1.7 dataset (Istomina et al. 2023) where we also observe the values as seen in this latest Istomina paper and we do not observe the step/decline present in V1.5.

We are very grateful to the reviewer for this suggestion, as by applying our pipeline to this V1.7 dataset also, we are not only able to successfully emulate melt pond fraction on the newer dataset, but our skill scores also improve. We explain this as being due to a lack of step change disparity between satellite products in this updated dataset. Furthermore, models here can be simpler which aids interpretability of melt pond processes and our physical understanding. We provide an example below for the V1.7 dataset: a simple emulator (red) versus observations (green), trained on V1.7 training data (not shown) and evaluated on the V1.7 test data portion (shown). We now include such examples in our manuscript.

[Figure]

We have therefore now noted in our paper the limitations of using the V1.5 dataset alongside the success of this approach by testing it on the latest V1.7 dataset. We furthermore note this highlights the fact that our approach and pipeline is robust and general across different datasets – valuable for future data releases.

We are very grateful to the reviewer and the extra work it led to. By showing success on the newer Istomina et al. (2023) V1.7 dataset, that the reviewer guided us to, we feel it has improved our manuscript substantially.

3. Results
Section 3 presents results in a structurally disorganized manner. A more coherent approach would be to present the model validation results first, followed by the test results.

We thank the reviewer for this useful comment. We have organised our results in a more structured manner which aids clarity and a better narrative structure.

The current presented results have at least two critical shortcomings: the performance and validation of the emulator. The authors described "the emulator shows a very strong similarity to the observed MPF" in line 184, however, from my perspective, although the emulator shows overall similarity with observations in the test set, there are many obvious mismatches (see areas circled in green in the figure below). Thus, the authors' description of the results is imprecise and overly colloquial. On the other hand, I believe the current validation approach is ineffective. It is necessary to compare the emulator with the original parameterization scheme on the same test set, rather than only comparing it against the climatology (lines 190-192). In other words, I believe the authors have not achieved their stated goal of replacing the physical parameterization scheme with the emulator (also refer to my fourth comment on the methods).

Furthermore, I suggest that the authors validate the emulator's effectiveness using MOSAiC data (e.g., Webster et al., 2022) and compare it with the original parameterization scheme, which would render the study more comprehensive.

[Figure]

Melt Pond Fraction: Emulator vs. Observations (with baseline)

We thank the reviewer for pointing this out. We have added extra discussion around the points the reviewer has highlighted. We also note that no emulator made from observations (which are noisy, sometimes missing and so on) will ever be a perfect replica of the physical phenomena and thus note the limitations they may have.

As to the second point, we apologise for the confusion caused which is a key misunderstanding due to our text. We do not involve a physical parametrisation in our paper, and it is not our aim to replace a physical parametrisation in our paper. We note that these developments could indeed aid data-driven components in the future. The direct goal of our paper is to show that a pointwise emulator can predict observed melt pond fraction from other observational variables and, therefore we do not have a physical parametrisation to work with here. We have clarified that this is a notable step for many areas in climate research which struggle to go from model data to observational data, and we have expanded on this value, but we are clearer now about the many challenges one might face before such methods are able to replace model parametrisation schemes, which is not our goal. We have hopefully clarified this in the text.

Other issues:

Please note that because there are numerous language and formatting issues in this manuscript, only several of them are listed below. To improve the quality of your manuscript, I recommend thoroughly revising the language to ensure a smoother flow and clarity.

We thank the reviewer for these insights and address both the specific comments below and have gone through this more thoroughly in general.

- The language of this paper is excessively verbose and lacks academic rigor. I mean, one should not use vague terms such as "very" to describe results (e.g., line 60, line 62, line 66, line 184, line 244).

Thank you for this, we have removed these terms and tightened our language. We have also gone through the text to make it sound more professional.

- Figure 1: Which line represents the "emulator" mentioned in the title?

This a typo. We apologise and have amended the title.

- Figure 3: I do not think this simple training workflow worth a schematic figure to illustrate. The only information I can get from this schematic figure is that the authors interpolated the features onto a widely used polar stereographic projection grid.

When presenting to audiences that were not familiar with machine learning they found this schematic helpful to visualise an ML training pipeline and process. In order to be inclusive to audiences we have kept the schematic for now, but we are happy to be flexible.

- Lines 187-189: I am unsure if the emulator has not seen this "large scale refreezing" in the test dataset, and thus, I am not convinced by this statement.

We agree that suggesting it is a large-scale refreezing is ambitious and have gone with the much more modest suggestion that it might capture physical events in the Arctic associated with weather variability.

- Lines 228-229: What does this sentence mean? Not clear.

We mean would anticipate an emulator trained on observations ideally then matches better the observed melt onset – because it is on that dataset on which it is essentially trained. Thus, we have added "For example, an emulator trained

on observations inherently captures the timing of melt onset as recorded in those observations". We have clarified this adding more detail.

- The formatting of the references is highly messy.

We thank the reviewer for pointing this out, some of which came from the bibtex. We have reformatted the references extensively to be neat and consistent.

References:

Feng, J., Zhang, Y., Cheng, Q., and Tsou, J. Y.: Pan-Arctic melt pond fraction trend, variability, and contribution to sea ice changes, Global and Planetary Change, 217, 103932, https://doi.org/10.1016/j.gloplacha.2022.103932, 2022. Istomina, L., Niehaus, H., and Spreen, G.: Updated Arctic melt pond fraction dataset and trends 2002–2023 using ENVISAT and Sentinel-3 remote sensing data, https://doi.org/10.5194/tc-2023-142, 22 September 2023.

Webster, M. A., Holland, M., Wright, N. C., Hendricks, S., Hutter, N., Itkin, P., Light, B., Linhardt, F., Perovich, D. K., Raphael, I. A., Smith, M. M., Von Albedyll, L., and Zhang, J.: Spatiotemporal evolution of melt ponds on Arctic sea ice,Elementa: Science of the Anthropocene, 10, 000072, https://doi.org/10.1525/elementa.2021.000072, 2022.

Xiong, C. and Ren, Y.: Arctic sea ice melt pond fraction in 2000–2021 derived by dynamic pixel spectral unmixing of MODIS images, ISPRS Journal of Photogrammetry and Remote Sensing, 197, 181–198, https://doi.org/10.1016/j.isprsjprs.2023.01.023, 2023.